# Neuropathic Pain Induces Interleukin-1β Sensitive Bimodal Glycinergic Activity in the Central Amygdala

**DOI:** 10.3390/ijms23137356

**Published:** 2022-07-01

**Authors:** Carolina A. Oliva, Jimmy Stehberg, Rafael Barra, Trinidad Mariqueo

**Affiliations:** 1Facultad de Educación, Universidad de Las Américas, República 71, Santiago 8370040, Chile; 2Laboratorio de Neurobiología, Instituto de Ciencias Biomédicas, Facultad de Medicina y Facultad de Ciencias de la Vida, Universidad Andrés Bello, República 330, Santiago 8370186, Chile; jstehberg@unab.cl; 3Centro de Investigación Biomédica y Aplicada (CIBAP), Escuela de Medicina, Facultad de Ciencias Médicas, Universidad de Santiago de Chile, Av. Libertador Bernardo O’Higgins 3677, Santiago 8320000, Chile; rafael.barra@usach.cl; 4Laboratorio de Neurofarmacología, Centro de Investigaciones Médicas, Facultad de Medicina, Universidad de Talca, Av. Lircay S/N, Talca 3460000, Chile

**Keywords:** neuropathic pain, glycine receptors, spontaneous inhibitory currents, CCI

## Abstract

Neuropathic pain reduces GABA and glycine receptor (GlyR)-mediated activity in spinal and supraspinal regions associated with pain processing. Interleukin-1β (IL-1β) alters Central Amygdala (CeA) excitability by reducing glycinergic inhibition in a mechanism that involves the auxiliary β-subunit of GlyR (βGlyR), which is highly expressed in this region. However, GlyR activity and its modulation by IL-1β in supraspinal brain regions under neuropathic pain have not been studied. We performed chronic constriction injury (CCI) of the sciatic nerve in male Sprague Dawley rats, a procedure that induces hind paw plantar hyperalgesia and neuropathic pain. Ten days later, the rats were euthanized, and their brains were sliced. Glycinergic spontaneous inhibitory currents (sIPSCs) were recorded in the CeA slices. The sIPSCs from CeA neurons of CCI animals show a bimodal amplitude distribution, different from the normal distribution in Sham animals, with small and large amplitudes of similar decay constants. The perfusion of IL-1β (10 ng/mL) in these slices reduced the amplitudes within the first five minutes, with a pronounced effect on the largest amplitudes. Our data support a possible role for CeA GlyRs in pain processing and in the neuroimmune modulation of pain perception.

## 1. Introduction

Neuropathic pain is a condition that involves a complex combination of physiological symptoms, including the sustained perception of pain [1,2]. The central nucleus of the amygdala (CeA), also called the ‘nociceptive’ amygdala, is involved in the cognitive and emotional integration of peripheral nociceptive pain signaling [3,4]. The CeA receives processed information from the basolateral-lateral (BLA-LA) amygdala and sends projections to the brainstem and forebrain, becoming the main amygdalar output to pain-associated modulatory systems [4].

During the establishment of neuropathic pain, the CeA shows increased excitability associated with reduced inhibitory control [5,6]. Current evidence shows that neuropathic pain induces increased spontaneous activity and persistent stimulus evoked-activity in CeA neurons [7]. This neuroplasticity appears to be caused by an imbalance between excitatory-inhibitory mechanisms rather than an increase in inputs from the noxious stimulus [4,7]. The inactivation of both the CeA and BLA reverses some characteristics of neuropathic pain, such as hyperalgesia, allodynia, and depressive-like behavior [8], while the use of inhibitory agonists in CeA rather than recovers, improves nociceptive behaviors [8].

Glycine is the primary inhibitory neurotransmitter in the ventral and dorsal horn of the spinal cord and brainstem [9]. Compared to GABA_A_, glycine is less abundant in the brain and presents an uneven distribution but has similar inhibitory properties [10]. Both GABA_A_ receptors and GlyR regulate neuronal excitability in regions that are critical for central nociceptive processing, including the spinal cord [11], thalamus [12], prefrontal cortex [13], nucleus accumbens [14], hippocampus [15], periaqueductal gray [16], and amygdala [17]. Both receptors are composed of α- and β-subunits; while the α-subunits included the neurotransmitter binding sites and form the pore, the βGlyR subunits are auxiliary modulatory subunits. Interestingly, the βGlyR subunit is highly expressed in the CeA, suggesting that GlyR may have a role in pain perception [17].

GlyRs are located at the postsynaptic membrane, anchored to gephyrin. This scaffold protein can modulate the balance towards more or less inhibitory strength, by regulating the number of available GlyRs at the membrane [18,19]. The expression of different GlyR subunits, channel conformation, localization in the synaptic and extrasynaptic sites, and allosteric modulation are strongly regulated in pathological states. Local or systemically released proinflammatory cytokines act as modulatory factors during neuropathic pain. In the dorsal horn, interleukin-1β (IL-1β) increases excitatory neurotransmission and reduces inhibition, by decreasing GABA- and glycine-mediated currents, leading to central pain sensitization and hypersensitivity [20]. Indeed, in spinal cord cultured neurons, the binding of IL-1β onto specific GlyR subunits has been reported, regulating the clustering and localization of GlyRs in the membrane [21]. However, no studies have so far evaluated glycinergic activity in supraspinal regions during neuropathic pain.

We recently reported that spontaneous glycinergic currents in CeA slices are modulated by IL-1β [22]. The described dual effect of IL-1β first increases glycinergic current amplitude by affecting GlyR channel opening and then reduces the amplitudes by disaggregating glycinergic clusters. This effect is caused by the interaction of IL-1β with at least one amino acid at the extracellular domain of βGlyR, a mechanism that could affect the availability of GlyR at the membrane, and how the amygdala perceives pain [22]. In the present work, we studied the glycinergic transmission in CeA slices and its modulation by IL-1β but in a condition of neuropathic pain induced by chronic constriction nerve injury of the sciatic nerve.

## 2. Results

### 2.1. Von Frey Mechanical Thresholds in the CCI Neuropathic Pain Model

To explore the role of glycinergic neurotransmission in the CeA during neuropathic pain, we performed chronic constriction of the sciatic nerve in adult Sprague Dawley male rats. Chronic constriction nerve injury (CCI) of the sciatic nerve is a model of neuropathic pain in which the sciatic nerve is loosely ligated to induce constriction, inducing neuropathic pain and hyperalgesia, which can be measured as allodynia (pain response to tactile stimulation) in response to hind paw plantar mechanical stimulation [23,24]. After the surgery, animals were allowed to recover before being tested with von Frey filaments to measure their mechanical withdrawal thresholds. The von Frey test measures the maximal pressure (measured in grams) exerted by the stimulation of the palm that triggers the animal to withdraw the paw. The animals exhibited a significantly reduced mechanical threshold after 3 and up to 10 days after CCI when compared to the Sham animals (at 3 days, Control: 34.9 ± 2.8 vs. Sham: 8.9 ± 2.1 g; at 10 days, Control: 34.7 ± 2.8 vs. Sham: 7.2 ± 2.3 g; n = 8, one-way ANOVA followed by the Bonferroni post hoc test *** *p* ˂ 0.001) (Figure 1). These data indicate that animals reduce their mechanical threshold by increasing sensitivity to pain, in a process known as allodynia.

### 2.2. Effects of CCI in Spontaneous Glycinergic Currents at the CeA

Ten days post-CCI surgery, the animals were euthanized, their brains extracted, cut into slices, and then whole-cell voltage-clamp recordings were performed in CeA neurons (Figure 2). We isolated and recorded the spontaneously generated glycinergic currents using a cocktail of ionic currents inhibitors. Data showed that Sham animals have a normal (unimodal) distribution of current amplitudes during spontaneous activity, with a peak at 16 pA (Figure 2A). Instead, animals that underwent CCI surgery displayed a bimodal distribution with two peaks, at 9 pA and 28 pA (Figure 2B). Representative current traces are displayed (Figure 2B, inset), while the main features of the amplitude distribution are included within the table (Figure 2C). The cumulative probability analysis (Figure 2D) showed that in CCI animals, low current amplitudes present a higher probability than in Sham animals; the opposite happens with the probability of higher amplitudes. Finally, we compared the mean values of glycinergic current amplitude, frequency, and decay time constant of the spontaneous events recorded (Figure 2E–G). The mean amplitude is significantly increased in the CCI group (Amplitude, Sham: 20.4 ± 0.2 pA vs. CCI: 21.8 ± 0.3 pA; *t*-test_(3.85, 9)_, ** *p* > 0.01, Figure 2E) but not the frequency (Sham: 0.15 ± 0.01 Hz vs. CCI: 0.18 ± 0.01 Hz, *p* = 0.105, Figure 2F), indicating that the postsynaptic GlyR but not the neurotransmitter release from presynaptic terminals is modified as an effect of neuropathic pain. The decay time constant also did not change (Sham: 0.72 ± 0.07 ms vs. CCI: 0.62 ± 0.04 ms; *p* = 0.265; Figure 2G), indicating that the small and large amplitudes correspond to the same type of ion channels. These data show that neuropathic pain modifies the amplitude of the postsynaptic GlyRs in the CeA.

### 2.3. IL-1β Modulates Spontaneous Glycinergic Currents in CCI Animals

To understand the physiological relevance of IL-1β modulation in neuropathic pain, we recorded glycinergic sIPSCs in CeA neurons from brain slices of CCI animals and perfused them with IL-1β 10 ng/mL (Figure 3). We plotted the current amplitude distribution for each condition and compared them at 5 (Figure 3A) and 15 min (Figure 3B) after the application of IL-1β. Representative traces of isolated spontaneous glycinergic currents from CeA neurons of CCI animals before and after IL-1β are displayed (Figure 3B, inset).

Notably, the perfusion with IL-1β almost completely eliminated the bimodal distribution within the first 5 min; the distribution became unimodal, with the lower amplitudes mainly represented. This new distribution lasted over time, with peaks at 11 pA and 14 pA after 5 min or 15 min of IL-1β application, respectively (Figure 3A,B). The table shows the main features of the amplitude current distribution of CeA neurons from CCI animals with and without perfusion with IL-1β (Figure 3C). The cumulative probability showed that bimodality lost power and that IL-1β increased the probability of the lower amplitude range compared to CCI alone (Figure 3D). We plotted the amplitude, frequency, and decay time constant mean values of spontaneous currents before and two times after IL-1β perfusion (Figure 3E–G). The data showed that IL-1β significantly reduced the amplitude of the sIPSCs (CCI: 21.8 ± 0.3 pA vs. +IL-1β_(5min)_: 14.2 ± 0.3 pA vs. +IL-1β_(15min)_: 18.7 ± 0.4 pA; one-way ANOVA followed by Bonferroni post-hoc test *** *p* < 0.001, Figure 3E). However, neither frequency (CCI: 0.18 ± 0.01 Hz vs. +IL-1β_(5min)_: 0.19 ± 0.03 Hz vs. +IL-1β_(15min)_: 0.17 ± 0.01 Hz; *p* = 0.64, Figure 3F) nor the decay time constant were significantly changed (CCI: 0.62 ± 0.04 ms vs. +IL-1β_(5min)_: 0.45 ± 0.05 Hz vs. +IL-1β_(15min)_: 0.53 ± 0.05 ms; *p* = 0.07, Figure 3G). These data shows that IL-1β strongly reduces the glycinergic current amplitudes, especially higher amplitudes, suggesting that it may be affecting the postsynaptic clustering structure of GlyRs.

## 3. Discussion

In the present work, we studied the spontaneous glycinergic currents in CeA slices from animals under neuropathic pain using the CCI neuropathic pain model. Our results show that CCI induces a bimodal distribution of spontaneous glycinergic current amplitudes, utterly different from the obtained from Sham animals. In normal conditions, the spontaneous glycinergic currents have a normal distribution. However, in peripheral pain, the currents are distributed in two populations, one lower amplitude peak at 9 pA, which likely represents unitary currents, as shown elsewhere [14], and a large amplitude peak at 28 pA, which may represent the conductance of three or more channels aggregated in clusters. Further studies will be needed to assess this hypothesis.

Our data show that neuropathic pain can alter ionic conductances allocated in supraspinal brain centers. In particular, the inhibitory conductance generated by GlyR in CeA appears to be persistently modulated. The increase in the average current amplitude found after CCI could be explained by the number of events of high amplitudes. The increased GlyR expression or increased channel permeability are not discarded, but given the amplitude distribution, the explanation of the higher amplitudes by clustering seems the most plausible. CCI does not affect other kinetic parameters like the frequency that depends on the released vesicles from presynaptic sites or the decay time that affects the time the channel is open.

In consequence, our results may point to the idea that neuropathic pain could induce aggregation into clusters of GlyRs in CeA interneurons. Clustering of GlyRs depends on interactions with the scaffolding protein gephyrin, modulated by different GlyRα and β subunits [25,26,27]. Further research should be made to determine if the bimodal current distribution and the high amplitude currents found after CCI are associated with the clustering of GlyRs, which could be estimated by colocalizing GlyRs with gephyrin.

Neuropathic pain induced by the CCI surgery, like other inflammatory processes, causes the increment of systemic inflammatory factors, such as proinflammatory cytokines (TNFα, IL-6, IL-1β), most of which can reach the central nervous system [20]. The cytokines can also be released in the brain by astrocytes and microglia [28]. We performed CCI surgery to generate a model of neuropathic pain by peripheral nerve injury. After ten days of CCI surgery, we observed a reduced mechanical threshold using the von Frey test, suggesting allodynia. The proinflammatory cytokine IL-1β is one of the most potent hyperalgesic agents released during peripheral trauma or nerve injury that increases the brain’s nociceptive signals [29]. Systemic IL-1β can increase excitatory currents and decrease inhibitory GABA and GlyR currents in the spinal cord [20]. However, one study showed that IL-1β induces potentiation of glycinergic currents in a specific spinal neuronal population, but through an indirect mechanism that involves the sequential binding of IL-1β to its receptor, the Ca^2+^ entry, and the increase of gephyrin-binding of GluRs at the membrane [26].

In a previous study, we reported that in CeA neurons, IL-1β induces a short-lasting increment in glycinergic currents five minutes after the addition of IL-1β. However, fifteen minutes later, the whole range of amplitudes diminished, suggesting that IL-1β disaggregates the GlyR clusters and induces endocytosis [22]. Here, in slices from CCI animals, we observed that IL-1β reduces the whole range of current amplitudes, and the bimodality is almost lost, with a shift of the peak amplitudes from the bimodal “2 peaks” at 9pA and 28pA, to 11 pA at 5 min and 14 pA, after 15 min of IL-1β incubation. The bimodality developed after CCI is interesting. We propose that the bimodal response observed after CCI is produced by alterations in the circuitry associated with pain within the amygdala. CCI could induce a new equilibrium different from Sham, with increased GlyR amplitudes in pain-related circuitry within the CeA. At the same time, the circuitries that are not involved in pain processing still could produce a broad amplitude spectrum of glycinergic currents. Hence, the type of glycinergic response may allow us to distinguish pain-related neuronal networks from other non-pain-associated CeA networks.

The external perfusion of IL-1β onto the CeA slices reduces all the current amplitudes, including large-amplitude events. If the CCI had produced ten days of a permanent inflammatory response mediated by the release of IL-1β at the CeA, we would have expected no large amplitude currents still present. Hence, it is unlikely that the bimodal response seen after CCI is, at least directly, produced by only the continuous local release of IL-1β. We speculate that a new equilibrium is established during those ten days that could include other proinflammatory cytokines and GABA conductances. Inhibitory conductances are still needed to keep the balance with excitation, therefore is a matter of determining how GABA currents change while glycinergic currents decrease by the action of IL-1β. More studies are required to test this hypothesis.

The central amygdala is one of the nuclei of the amygdaloid complex, associated with the emotional processing and perception of pain [4,5,8,27]. CeA receives pain signals from the spinal cord through the parabrachial nucleus (PBN) via the spino-parabrachial pathway [3,4]. Interestingly, the stimulation of the PBN-CeA pathway is enough to drive behaviors of negative emotions. In contrast, the top-down activity from BLA-CeA counteracts those adverse effects, regulating anxiety behavior and having an anti-depressive effect [3]. This control of negative emotions might change when pain turns from acute to chronic due to adaptations in activity-related stimuli that surpass the control of aversive emotions mentioned above [3]. Together with individual vulnerabilities, a condition of permanent pain that activates PBN-CeA could trigger the development of psychiatric disorders and chronic diseases [3,27].

Surprisingly, CeA contains unusually high expression levels of the βGlyR subunit, the modulatory auxiliary subunit of GlyR, compared to other brain regions [17]. There is no clarity about the high presence of this subunit and if it is related to some function. Glycine is the second inhibitory conductance in the central nervous system, with an uneven location. We suggest that the βGlyR subunit is part of a neuroimmune modulation mechanism that can modulate emotional perception through its interaction with IL-1β. Given the function of CeA described above, we suggest the βGlyR subunit could be part of the PBN-CeA or BLA-CeA pathways, modulating the individual experience perception during neuropathic pain. By the modulation of IL-1β, spontaneous GlyR currents can increase or decrease, integrating the signals in or out of the CeA. Hence, the type of glycinergic response may allow us to distinguish the involvement of these pathways in neuropathic disease, a hypothesis that remains to be solved.

In the present work, we evidenced the state of GlyR currents in CeA under a neuropathic pain condition. Although the perfusion of IL-1β can explain the acute effect on GlyR currents, a persistent condition such as neuropathic pain would better resemble chronic inflammation’s neurological status. The pain sensation caused by nerve injury lasts long and causes detrimental effects on quality of life and emotional aspects [8,25,27]. The reduction in spinal glycinergic transmission contributes to generating hyperalgesia and allodynia during chronic neuropathic pain [25,28]. Our findings suggest that neuropathic pain changes glycinergic activity, which may affect the signal output from CeA; meanwhile, IL-1β modulates the integrated summation and output to and from CeA, and thus the emotional perception of neuropathic pain. How these changes in GlyRs affect pain perception per se or contribute to the development of cognitive deficits, anxiety disorders, and depression commonly associated with chronic pain and involve the amygdala are exciting questions that arise from the present study.

## 4. Materials and Methods

### 4.1. Animals

Male Sprague-Dawley (SD) rats (250–350 g) were obtained from the Animal Facility of the University of Chile. Every effort was made to minimize animal suffering. Animals were bred and housed in controlled laboratory conditions, received a standard rat chow diet and water ad libitum, and were housed on a 12-h light/dark cycle at a constant room temperature of 23 °C. Animal care and experimental protocols for this study were approved by the Institutional Animal Use Committee of University of Chile (N°17027-MED-UCH) and of Universidad Andrés Bello (N°11-2016).

### 4.2. Chronic Constriction Injury (CCI)

The sciatic chronic constriction nerve injury in Sprague Dawley rat is a validated model for the study of neuropathic pain [23]. Briefly, the rats were anesthetized with ketamine/xylazine (100/10 mg/kg), and the dissection through the right biceps femoris was performed to expose the sciatic nerve. Four ligatures (chromic catgut 4-0) were tied, and then the incision was closed by layers. In the control (Sham) condition, all the procedures were the same as CCI without the ligation. In all the procedures, we ensured the appropriate aseptic techniques were used. The animals were maintained on a temperature-controlled pad until they recovered from the anesthesia, were given tramadol (100 mg/kg) for treatment of post-surgical pain, and monitored throughout the study for their wellbeing and non-CCI-related abnormalities (e.g., excessive loss of weight, chronic piloerection, belly swelling, extended periods of immobilization and appearance of sickness). No animals met the exclusion criteria defined by the Morton and Griffiths guidelines [30], and hence, they were all included in the analysis.

### 4.3. Von Frey Mechanical Threshold

Rats were placed into individual chambers with wire mesh floors and transparent covers. After 15 min of habituation, their mechanical threshold was manually evaluated using the von Frey filaments. A perpendicular force was applied to the mid-plantar zone of each paw using filaments that reached a pre-determined pressure (32–512 mN) before bending, ordered from thinner to thicker until a withdrawal response was elicited. The mechanical withdrawal threshold was determined using the Up-Down method with three up-down cycles per measure [31].

### 4.4. Electrophysiological Recordings

For electrophysiological recordings, male rats that underwent CCI or control (Sham) surgery were used 10 days after the surgery, and the treatment that each animal underwent was not known to the person who performed the recordings. Every day, one animal was anesthetized with isoflurane and then decapitated. The brain was quickly removed, and placed in a beaker with cold artificial cerebrospinal artificial solution (ACSF)-modified with sucrose, composed of (in mM): 85 NaCl, 75 sucrose, 3 KCl, 1.25 NaH_2_PO_4_, 25 NaHCO_3_, 10 dextrose, 3.5 MgSO_4_, 0.5 CaCl_2_, 3 sodium pyruvate, 0.5 sodium L-ascorbate and 3 Myo-inositol (305 mOsm, pH 7.4 with 95% O_2_/5% CO_2_). Coronal slices were cut (300 µm thick) with a vibratome. After 1 h at 36 °C, the slices were changed to ‘recording solution’, composed of (in mM): 126 NaCl, 3.5 KCl, 1.25 NaH_2_PO_4_, 25 NaHCO_3_, 10 dextrose, 1 MgSO_4_, 2 CaCl_2_, 3 sodium pyruvate, 0.5 sodium L-ascorbate and 3 Myo-inositol (305 mOsm, pH 7.4 with 95% O_2_/5% CO_2_) at room temperature (22 °C). Slices were recorded in a submerged-style chamber solution with ACSF at 30–32 °C under an upright infrared-differential interference contrast (IR-DIC) fluorescence microscope (Eclipse FNI, Nikon, Melville, NY, USA) equipped with a 40X water objective. We performed voltage-clamp whole-cell recordings in the soma of visually identified neurons allocated in the central amygdala. We used a borosilicate glass electrode (World Precision Instruments, Sarasota, FL, USA) pulled on a P-97 Flaming/Brown Micropipette Puller (Sutter Instruments, Novato, CA, USA). The glass pipette, ranging from 3.5 to 4.2 MΩ, was filled with intracellular solution containing the following (in mM): 130 Cs-gluconate, 3.5 CsCl, 4 ATP-Mg, 0.3 GTP-Na, 10 Na-phosphocreatine, 1 EGTA, 10 HEPES, and 0.4% Biocytin (286 mOsm, pH 7.4 adjusted with CsOH). After seal formation and successful transition to whole-cell configuration, access resistance was usually between 4–15 MΩ and series resistance was continuously monitored and compensated between 75–80%. We isolated glycine-mediated currents in the presence of synaptic blockers CNQX (10 μM), APV (50 μM), and bicuculline (10 μM) to block AMPAR, NMDAR, and GABA_A_ receptors, respectively. We recorded the spontaneous inhibitory postsynaptic currents (sIPSCs) obtained as outward currents at a holding potential of +20 mV. The remaining sIPSCs obtained after further application of bicuculline (10 µM) were considered glycine-based sIPSC. At the end of the recording, we applied strychnine to block glycinergic currents. We used PClamp detection events to choose single events of about 3 to 10 pA, showing greater than peak-to-peak noise to analyze the area and decay constant. Spontaneous synaptic events were detected over 100 s of continuous recording two times after the application of IL-1β (10 ng/mL). The signals for both voltage-clamps were acquired using a MultiClamp 700B amplifier (Axon CNS, Molecular Devices, San José, CA, USA), low-pass filtered at 10 kHz, digitally sampled at 30 kHz, and recorded through a Digidata-1440A interface (Axon CNS, Molecular Devices, San José, CA, USA) and PClamp 10.3 software. The analysis was performed from the recordings of 9 cells from 5 Sham rats and 10 cells from 5 CCI rats. In most cases, 2 cells per animal (one cell per slice) were recorded. Slices from each animal were considered replicates and averaged together. As the CeA is a small brain region, only 2 slices per hemisphere were obtained per animal.

### 4.5. Statistical Analysis

The results were expressed as the mean ± standard error (SEM). We used a *t*-test when two groups were analyzed and a one-way analysis of variance (ANOVA) when more than two groups were compared, followed by the Bonferroni’s post hoc test. The threshold for statistical significance was *p* < 0.05, with a 95% confidence interval (CI). Statistical analyses were performed using Prism software (GraphPad, San Diego, CA, USA).

## 5. Conclusions

In the model of neuropathic pain induced by CCI, the spontaneous glycinergic currents in the CeA are distributed in two populations of small and large current amplitudes. The exposure to IL-1β reduces both populations’ amplitudes but with a more substantial effect on the population with larger current amplitudes.

## Figures and Tables

**Figure 1 ijms-23-07356-f001:**
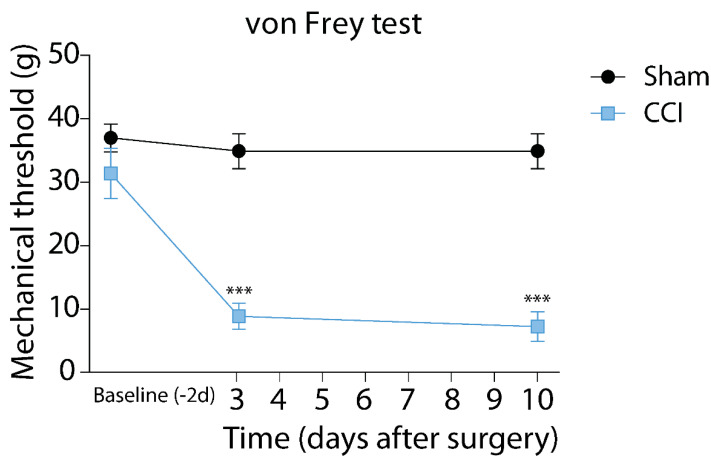
Von Frey mechanical thresholds in the CCI neuropathic pain model. *** *p* < 0.001.

**Figure 2 ijms-23-07356-f002:**
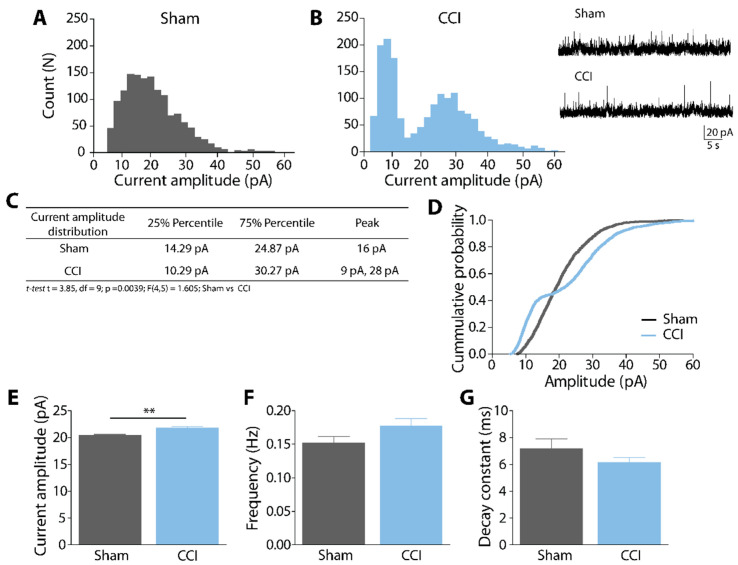
Glycinergic currents (sIPSCs) in CeA slices of CCI show a bimodal current amplitude distribution. (**A**,**B**) Spontaneous current amplitude distribution plots for Sham (**A**) and CCI (**B**) animals. Inset: representative synaptic sIPSC glycinergic currents in the presence of CNQX (10 μM), APV (50 μM), bicuculline (10 μM), clamped at +20 mV. (**C**) the table summarizes the distribution data from (**A**,**B**). (**D**) Cumulative probability histogram of current amplitude in CeA neurons in the Sham and CCI. (**E**–**G**) Comparative plots of the glycinergic sIPSC current parameters, as mean ± SEM, ** *p* < 0.01 (n = 5 Sham, 9 cells; n = 5 CCI, 10 cells).

**Figure 3 ijms-23-07356-f003:**
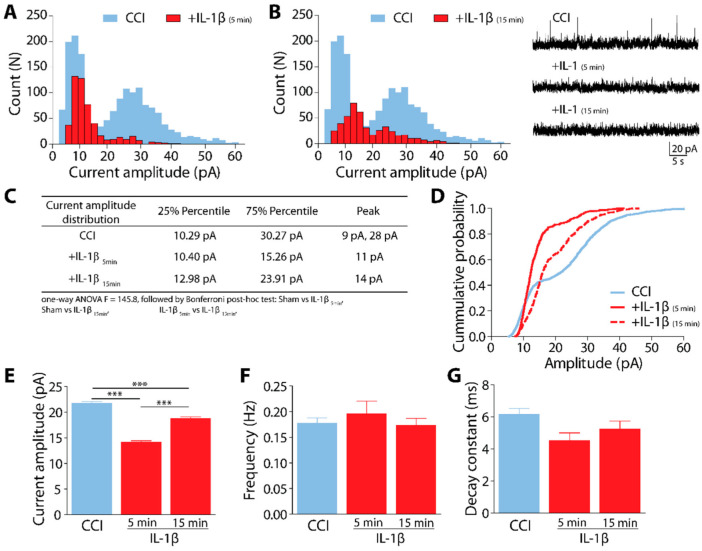
Glycinergic sIPSCs in CeA slices from CCI animals are differentially modulated by IL-1β. (**A**,**B**) Spontaneous current amplitude distribution plots for CCI slices, before and after 5 min (**A**) or 15 min (**B**) IL-1β (10 ng/mL). Inset: representative synaptic sIPSC glycinergic currents in the presence of CNQX (10 μM), APV (50 μM), bicuculline (10 μM), clamped at +20 mV. (**C**) the table summarizes the distribution data from (**A**,**B**). (**D**) Cumulative probability histogram of amplitude in CeA neurons in CCI, before (blue line) and after 5 min (continuous red line) and 15 min (discontinuous red line) of IL-1β. (**E**–**G**) Comparative plots of the glycinergic sIPSC currents parameters, as mean ± SEM, *** *p* < 0.001 (n = 5 CCI, 10 cells; 10 cells + IL-1β).

## Data Availability

All data sets are included in the manuscript.

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
