# Peer review of "Neuropathic Pain Induces Interleukin-1β Sensitive Bimodal Glycinergic Activity in the Central Amygdala"

_ijms, 2022, doi:10.3390/ijms23137356_

Round 1
Reviewer 1 Report
Thank you for permitting me to review this manuscript
Introduction
Line 76 please clearly state the primary and secondary objective of the present study
Line121
Previous work determined that spontaneous GlyR currents in CeA were modulated by IL- 121 1β [22]. This phrase would better be transfered to the introduction section
Line 196_ 209
Pleas adjust the police size and formatting as characters appears smaller
Line 322
conclusion should only cite facts issued from the study and suggest , line 323 324 should be deleted as the word perhaps is used , of course it can be added in the dioscussion section in association with possible clinical outcomes of the findings
chronic pain is a condition which last 3 months in human adults , how this experience cope with this issue , the authors should explain why their model is chronic pain rather than acute pain
the authors presents 30 references but cites 32 , indeed ref 31 is supposed to be the description of their model for chronic pain please correct and elaborate more your model of chronic pain may be by more references
Author Response
Dear Editor:
We wish to thank you and the Reviewers 1 and 2, for your time and comments, which are greatly appreciated. No doubt they have helped us improve the Manuscript substantially. The comments of the Reviewers are each followed by a response, followed by an excerpt of the changes in the Manuscript to ease their revision of the changes. We have also highlighted within the Manuscript the changes, in yellow for those changes associated to the comments of Reviewer 1 and in green for Reviewer 2.
We hope that the reviewers are satisfied with the corrections made. We hope that we have been able to adequately address all Reviewers´ comments and that the Manuscript will be now deemed suitable for publication.
Comments to the Reviewer #1: (highlighted in yellow)
Introduction
- Reviewer: Line 76 please clearly state the primary and secondary objective of the present study
Response: Thank you for this comment. We have modified the text as follows:
“In the present work, we studied glycinergic transmission in CeA slices and its modulation by IL-1b but in a condition of neuropathic pain induced by chronic constriction nerve injury.”
Line121
- Reviewer: Previous work determined that spontaneous GlyR currents in CeA were modulated by IL- 121 1β [22]. This phrase would better be transferred to the introduction section
Response: Thanks for the suggestion. We relocated this sentence; now it is the first sentence of the introduction’s last paragraph. It reads:
“We recently reported that spontaneous glycinergic currents in CeA slices are modulated by IL-1β [22].”
- Reviewer: Line 196_ 209. Please adjust the police size and formatting as characters appears smaller
Response: Thanks for noticing; corrected.
- Reviewer: Line 322, conclusion should only cite facts issued from the study and suggest, line 323 324 should be deleted as the word perhaps is used, of course it can be added in the discussion section in association with possible clinical outcomes of the findings.
Response: Thank you for this comment. We relocated this sentence to the last paragraph of the discussion and rewrote the conclusion as follows:
"In the model of neuropathic pain induced by CCI, the spontaneous glycinergic currents in the CeA are distributed in two populations of small and large current amplitudes. The exposure to IL-1β reduces both populations' amplitudes but with a more substantial effect on the population with larger current amplitudes."
- Reviewer: Chronic pain is a condition which last 3 months in human adults, how this experience cope with this issue, the authors should explain why their model is chronic pain rather than acute pain
Response: We understand the point raised by the Reviewer. The present work describes the modulation of glycinergic currents by IL-1b in the CCI model of neuropathic pain. Our study was evaluated ten days post-CCI because it was reported that GlyRs subunit expression changes in the spinal cord after this period of time post-CCI surgery (Esmaeili & Zaker, 2011. Differential expression of glycine receptor subunit messenger RNA in the rat following spinal cord injury. Spinal Cord. 49(2):280–284. doi: 10.1038/sc.2010.109). Hence, the use of the term “chronic” was indeed unnecessary and misleading. We changed the term “chronic” for “neuropathic” throughout the Manuscript.
- Reviewer: The authors present 30 references but cites 32, indeed ref 31 is supposed to be the description of their model for chronic pain please correct and elaborate more your model of chronic pain may be by more references
Response: Thank you for noticing this mistake; we have corrected it in the new version of the manuscript.
Reviewer 2 Report
This is an interesting article proposing a novel CeA mechanism underlying neuropathic pain in the rat CCI model. While interesting, some major flows are contained in the study. Please see my comments below:
Abstract:
The abstract needs rewriting. There are some sentences that make no sense (i.e. line 20: this sentence is missing a verb) and one should consider specifying on who the CCI was performed, and that the animals were euthanized and brain slices analyzed ex vivo. Also 10 days of CCI; can this be defined as chronic pain?? what do you mean by "the application of IL-1b in CCI": injection to the mice, experiment ex vivo? what is CCI defined as? Also the conclusions drawn in the abstract are by far too big for the experiments performed and should be toned down
The abstract, arguably the most important part of a manuscript, does not stand alone and does not accurately explain the purpose or experiments included in the paper. Please revise.
Intro:
- line 48- rather than recovers, improves nociceptive behaviours.
- line 51 -52 : is less abundant in the brain AND present and uneven distribution..
-line 61 : towardS
Results/Methods:
- were any animals excluded in the CCI group, or was there a predefined PWT at which they would have been excluded?
- Please report animal experiments according to the ARRIVE guidelines. No ethical permits for aniamle xperiments are cited in this article.
Authors state "The sciatic chronic constriction nerve injury in Sprague Dawley rat is a validated 262 model for the study of chronic pain [31]": is that right?? I believe is valdiated for neuropathic pain, and if maintained for long periods of time for chronic pain but 10 days can hardly be considered "long term". I struggle to believe that the CCI model reflects chronic pain on its own (i.e. if animals are exposed to the CCI surgery and then euthanized within 24 hours: how would the chronic pain mechanisms have time to settle down??!) and I would have liked to check the reference provided, but unfortunately it is not in the reference list, which was truncated at 30 refs.
-Importantly: what analgesics were provided to control post-surgical pain in these animals? In this day an age, post-surgical analgesia is an absolute requirement to maximize animal health and werlfare.
- And how was the von frey pressure maintaiend at those specific newtons if the filaments were applied manually? how was this recorded?
-Line 95: would be benefitial to mention that animals were euthanized prior to brain isolation.
-Line 111: I do not understand what the authors mean by "long lasting"?
- How is n calculated in figure 2? 5 sham rats, 9 cells from each of 9 cells in total? are these cells plotted individually or is the mean of all cells pertaining one rat showed
Discussion/Conclusion:
- chronic pain after just 10 days?
- I believe that further spatial confirmation of whether GlyRs cluster in CeA neurons may be needed to support the statements made by the authors.
- should one consider analyzing anxiety-like behavuours in these animals, or other behavioural tests that assess the "pain" rather thana llodynia (such as CPP)?
Author Response
Dear Editor:
We wish to thank you and the Reviewers 1 and 2, for your time and comments, which are greatly appreciated. No doubt they have helped us improve the Manuscript substantially. The comments of the Reviewers are each followed by a response, followed by an excerpt of the changes in the Manuscript to ease their revision of the changes. We have also highlighted within the Manuscript the changes, in yellow for those changes associated to the comments of Reviewer 1 and in green for Reviewer 2.
We hope that the reviewers are satisfied with the corrections made. We hope that we have been able to adequately address all Reviewers´ comments and that the Manuscript will be now deemed suitable for publication.
Comments to the Reviewer #2: (highlighted in green)
Abstract:
- Reviewer: The abstract needs rewriting. There are some sentences that make no sense (i.e. line 20: this sentence is missing a verb) and one should consider specifying on who the CCI was performed, and that the animals were euthanized and brain slices analyzed ex vivo.
Response: We wish to thank the reviewer for this comment. We rephrased the abstract in order to make all the sentences as clear as possible. Now it reads as follows:
“Neuropathic pain reduces GABA and glycine receptor (GlyR)-mediated activity in spinal and supraspinal regions associated with pain processing. Interleukin-1β (IL-1β) alters Central Amygdala (CeA) excitability by reducing glycinergic inhibition in a mechanism that involves the auxiliary β-subunit of GlyR (βGlyR), which is highly expressed in this region. However, GlyR activity and its modulation by IL-1β in suprspinal brain regions under neuropathic pain have not been studied. We performed chronic constriction injury (CCI) of the sciatic nerve in male Sprague Dawley rats, a procedure that induces hindpaw plantar hyperalgesia and neuropathic pain. Ten days later, the rats were euthanized, and their brains were sliced. Glycinergic spontaneous inhibitory currents (sIPSCs) were recorded in the CeA slices. The sIPSCs from CeA neurons of CCI animals show a bimodal amplitude distribution, different from the normal distribution in Sham animals, with small and large amplitudes of similar decay constants. The perfusion of IL-1β (10 ng/mL) in these slices reduced the amplitudes within the first 5 min, with a pronounced effect on the largest amplitudes. Our data supports a possible role for CeA GlyRs in pain processing and in the neuroimmune modulation of pain perception.”
- Reviewer: Also 10 days of CCI; can this be defined as chronic pain??
Response: This same issue was raised by Reviewer 1. The present work describes the modulation of glycinergic currents by IL-1b in the CCI model of neuropathic pain. Our study was evaluated ten days post-CCI because it was reported that GlyRs subunit expression changes in the spinal cord after this period of time post-CCI surgery (Esmaeili & Zaker, 2011. Differential expression of glycine receptor subunit messenger RNA in the rat following spinal cord injury. Spinal Cord. 49(2):280–284. doi: 10.1038/sc.2010.109). Hence, the use of the term “chronic” was indeed unnecessary and misleading. We changed the term “chronic” for “neuropathic” throughout the Manuscript.
- Reviewer: What do you mean by "the application of IL-1b in CCI": injection to the mice, experiment ex vivo?
Response: We apologize for this. It meant “perfusion of IL-1β on slices from CCI animals”. We rewrote that sentence.
- Reviewer: What is CCI defined as?
Response: Thank you for noticing that the CCI model had not been defined in the Manuscript. Given that the materials and methods section is located at the end of the Manuscript, we decided to add a paragraph defining CCI in the beginning of the results section. It now reads as follows:
“Chronic constriction nerve injury (CCI) of the sciatic nerve is a model of neuropathic pain in which the sciatic nerve is loosely ligated to induce constriction, eliciting neuropathic pain and hyperalgesia, which can be measured as allodynia (pain response to tactile stimulation) in response to hindpaw plantar mechanical stimulation [23,24].”
- Reviewer: Also the conclusions drawn in the abstract are by far too big for the experiments performed and should be toned down.
Response: We rewrote the abstract, toning down the conclusions.
- Reviewer: The abstract, arguably the most important part of a manuscript, does not stand alone and does not accurately explain the purpose or experiments included in the paper. Please revise.
Response: We appreciate the recommendations of the Reviewer. We revised the abstract considering the various recommendations of the Reviewer. It now reads:
“Neuropathic pain reduces GABA and glycine receptor (GlyR)-mediated activity in spinal and supraspinal regions associated with pain processing. Interleukin-1β (IL-1β) alters Central Amygdala (CeA) excitability by reducing glycinergic inhibition in a mechanism that involves the auxiliary β-subunit of GlyR (βGlyR), which is highly expressed in this region. However, GlyR activity and its modulation by IL-1β in suprspinal brain regions under neuropathic pain have not been studied. We performed chronic constriction injury (CCI) of the sciatic nerve in male Sprague Dawley rats, a procedure that induces hindpaw plantar hyperalgesia and neuropathic pain. Ten days later, the rats were euthanized, and their brains were sliced. Glycinergic spontaneous inhibitory currents (sIPSCs) were recorded in the CeA slices. The sIPSCs from CeA neurons of CCI animals show a bimodal amplitude distribution, different from the normal distribution in Sham animals, with small and large amplitudes of similar decay constants. The perfusion of IL-1β (10 ng/mL) in these slices reduced the amplitudes within the first 5 min, with a pronounced effect on the largest amplitudes. Our data supports a possible role for CeA GlyRs in pain processing and in the neuroimmune modulation of pain perception.”
Intro:
- Reviewer: - line 48- rather than recovers, improves nociceptive behaviours.
Response: We corrected the sentence. It now reads:
“, while the use of inhibitory agonists in the CeA improves nociceptive behaviors in a rat model of neuropathic pain [8].”
- Reviewer: - line 51 -52 : is less abundant in the brain AND present and uneven distribution..
Response: We corrected the sentence as recommended.
Reviewer: -line 61 : towards
Response: Corrected.
Results/Methods:
- Reviewer: - were any animals excluded in the CCI group, or was there a predefined PWT at which they would have been excluded?
Response: No animals were excluded from the analysis. In general, we would have excluded animals that met the exclusion criteria defined in the guidelines on the recognition of pain, discomfort and distress, by Morton and Griffits (Vet. Rec. 116:431-436, 1985), based on animals´ wellbeing and behavior, but as there were no animals meeting such criteria, no animal was excluded from the analysis.
We have added a sentence at the materials and methods section dealing with this issue, which now reads:
“No animals met the exclusion criteria defined by the Morton and Griffiths guidelines [29], and hence, they were all included in the analysis.”
- Reviewer: - Please report animal experiments according to the ARRIVE guidelines. No ethical permits for animal’ experiments are cited in this article.
Response: We thank the Reviewer for this comment and for noticing that the Bioethical protocol was not included. We added the approved protocol, and we also went through the ARRIVE guidelines to include all the data missing (obtained from https://arriveguidelines.org/resources/author-checklists). Based on the list, we have added statements about randomization and blinding in the materials and methods section and strain in the abstract.
The following sentence was added to the materials and methods section, mentioning the animal protocol approvals:
“Animal care and experimental protocols for this study were approved by the Institutional Animal Use Committee of University of Chile (N°17027-MED-UCH) and of Universidad Andrés Bello (N° 11-2016)”.
The following sentence was added regarding randomization in selecting the animals for the two treatment groups:
“Briefly, rats were randomized into groups that underwent either CCI or control (sham) surgery.”
The following sentence was added regarding blinding of the person who performed the electrophysiological recordings:
“For electrophysiological recordings, male rats that underwent CCI or control (sham) surgery were used 10 days after the surgery, and the treatment that underwent each animal was blinded to the person who performed the recordings”.
- Reviewer: Authors state "The sciatic chronic constriction nerve injury in Sprague Dawley rat is a validated 262 model for the study of chronic pain [31]": is that right?? I believe is validated for neuropathic pain, and if maintained for long periods of time for chronic pain but 10 days can hardly be considered "long term". I struggle to believe that the CCI model reflects chronic pain on its own (i.e. if animals are exposed to the CCI surgery and then euthanized within 24 hours: how would the chronic pain mechanisms have time to settle down??!) and I would have liked to check the reference provided, but unfortunately it is not in the reference list, which was truncated at 30 refs.
Response: As stated earlier, we agree with the Reviewer that CCI is a neuropathic pain model and 10 days may not be considered chronic. In consequence, we replaced the term “chronic” for “neuropathic” throughout the Manuscript.
The references list and in-text citations were checked and corrected.
- Reviewer: -Importantly: what analgesics were provided to control post-surgical pain in these animals? In this day an age, post-surgical analgesia is an absolute requirement to maximize animal health and welfare
Response: We appreciate this comment. To reduce post-surgical pain the animals were administered tramadol (100 mg/Kg). We have added a paragraph regarding post-surgical care. The paragraph reads as follows:
“The animals were maintained on a temperature-controlled pad until they recovered from the anesthesia, were given tramadol (100mg/kg) for treatment of post-surgical pain and monitored throughout the study for their wellbeing and for non CCI-related abnormalities (e.g. excessive loss of weight, chronic piloerection, belly swelling, extended periods of immobilization and appearance of sickness).”
- - And how was the von frey pressure maintained at those specific newtons if the filaments were applied manually? how was this recorded?
Response: We again apologize for the ambiguities in the text and thank the Reviewer for noticing them. Indeed, the pressure of the von frey filaments is exerted on the plantar surface until the maximal pressure is reached and the filament bends. In consequence, the pressure is not maintained. We have reworded the relevant sentence in the methods section into the following sentence:
“A perpendicular force was applied to the mid-plantar zone of each paw using filaments that reached a pre-determined pressure (32-512 mN) before bending, ordered from thinner to thicker, until a withdrawal response was elicited.”
- Reviewer: -Line 95: would be beneficial to mention that animals were euthanized prior to brain isolation.
Response: The sentence in question was changed as follows:
“Ten days post-CCI surgery, the animals were euthanized, their brains extracted, cut into slices, and then whole-cell voltage-clamp recordings were performed in CeA neurons”
- Reviewer: -Line 111: I do not understand what the authors mean by "long lasting"?
Response: We re-wrote this sentence for clarity about the results and the conclusion of this part: “The mean amplitude is significantly increased in the CCI group (Amplitude, sham: 20.4 ± 0.2 pA vs CCI: 21.8 ± 0.3 pA; t-test(3.85, 9), **p > 0.01, Figure 2E) but not the frequency (sham: 0.15 ± 0.01 Hz vs CCI: 0.18 ± 0.01 Hz, p = 0.105, Figure 2F), indicating that the postsynaptic GlyR but not the neurotransmitter release from presynaptic terminals is modified as an effect of neuropathic pain. The decay time constant also did not change (sham: 0.72 ± 0.07 ms vs CCI: 0.62 ± 0.04 ms; p = 0.265; Figure 2G), demonstrating that the small and large amplitudes correspond to the same type of ion channels. These data show that neuropathic pain modifies the amplitude of the postsynaptic GlyRs in the CeA.”
- Reviewer: How is n calculated in figure 2? 5 sham rats, 9 cells from each of 9cells in total? are these cells plotted individually or is the mean of all cells pertaining one rat showed
Response: The analysis was built out of 9 cells from 5 Sham rats and 10 cells from 5 CCI rats. In most cases, 2 cells per animal (one cell per slice) were recorded. Slices from each animal were considered replicates and averaged together. As the CeA is a small brain region, only 1 or 2 slices per hemisphere can be obtained per animal.
We added a short paragraph at the end of the electrophysiological recordings subsection of materials and methods dealing with this issue. It reads:
“The analysis was performed from the recordings of 9 cells from 5 Sham rats and 10 cells from 5 CCI rats. In most cases, 2 cells per animal (one cell per slice) were record-ed. Slices from each animal were considered replicates and averaged together. As the CeA is a small brain region, only 2 slices per hemisphere were obtained per animal.”
Discussion/Conclusion:
- Reviewer: chronic pain after just 10 days?
Response: As stated earlier, we replaced the term “chronic” for neuropathic” throughout the Manuscript.
- Reviewer: I believe that further spatial confirmation of whether GlyRs cluster in CeA neurons may be needed to support the statements made by the authors.
Response: We agree with the Reviewer that concluding that GluRs cluster in CeA neurons needs further experimental confirmation. Thus, we were careful to only suggest clustering based on our previous studies. The suggestion for GlyR clustering in the CeA comes from our previous study (Solorza et al., 2021), in which we suggested that the reduction in GlyR currents was due to GlyR de-clustering. The clustering of GlyRs is regulated by the interaction of the GlyR β-subunit with gephyrin (Maric et al., 2011). Using molecular docking and other techniques, we provided evidence that IL-1β binds to the GlyR β-subunit, arrangement closes the channel and probably induces its endocytosis (Huang et al., 2017). In the current Manuscript, we show that the smaller amplitude is three times shorter than the larger amplitudes, suggesting that the latter could represent aggregates of at least three channels. Still, more studies have to be made to confirm this point.
We have toned down this, clarifying, when possible, that concluding anything about clustering would require further studies. The relevant paragraphs in the discussion section read as follows:
“…which may represent the conductance of three or more channels aggregated in clusters. Further studies will be needed to assess this hypothesis.”
“In consequence, our results may point to the idea that neuropathic pain could induce the aggregation into clusters of GlyRs in CeA interneurons. Clustering of GlyRs depends on interactions with the scaffolding protein gephyrin, modulated by different GlyRα and β subunits [23–25]. Further research should be made to determine if the bimodal current distribution and the high amplitude currents found after CCI are associated with the clustering of GlyRs, which could be estimated by colocalizing GlyRs with gephyrin.”
Reviewer: should one consider analyzing anxiety-like behaviors in these animals, or other behavioral tests that assess the "pain" rather than allodynia (such as CPP)?
Response: This is a very interesting issue raised by the Reviewer. The CeA is involved in the orchestration of many behaviors, including anxiety-like behavior and even non-declarative memory (classical conditioning paradigms, such as cued fear conditioning). However, glycine by itself does not only act on GlyRs, but it is also a co-agonist of glutamatergic NMDAR receptors. In the electrophysiological recordings we isolate and measure GlyR activity, by blocking pharmacologically other receptors, including NMDARs. However, to test the role of CeA GlyRs in anxiety or memory in vivo, it is required to both activate and inhibit GlyRs selectively, as CeA NMDARs are involved in both anxiety-like behaviors and memory. Moreover, the selective inhibition of different GlyR subunits could also affect behavior and would be an interesting twist. In consequence, we are now performing experiments for a different Manuscript, which will focus specifically on this issue, by evaluating the effect of blocking and activating both GlyR and specific GlyR subunits in vivo at the CEA, in a behavioral battery, including responses to tactile stimuli, heat pain thresholds, hyperalgesia due to acute inflammatory pain (e.g. formalin, carrageenan), anxiety (EPM) and auditory fear conditioning.

Round 2
Reviewer 1 Report
The authors have responded to my concerns and have improved the manuscript